# SUNi mutagenesis: Scalable and uniform nicking for efficient generation of variant libraries

Taylor L. Mighell[1]*, Ignasi Toledano[1], Ben Lehner [1,2,3,4]*

**1** The Barcelona Institute of Science and Technology, Center for Genomic Regulation (CRG), Barcelona, Spain, **2** Universitat Pompeu Fabra (UPF), Barcelona, Spain, **3** Institució Catalana de Recerca i Estudis Avançats (ICREA), Barcelona, Spain, **4** Wellcome Sanger Institute, Wellcome Genome Campus, Hinxton, United Kingdom

* taylor.mighell@crg.eu (TLM); bl11@sanger.ac.uk (BL)

**Data Availability Statement:** Raw sequencing data produced for this study can be found at the Sequence Read Archive with accession number PRJNA939024.

**Funding:** This work was funded by a European Research Council (ERC) Advanced grant (883742),

## Abstract

Multiplexed assays of variant effects (MAVEs) have made possible the functional assessment of all possible mutations to genes and regulatory sequences. A core pillar of the approach is generation of variant libraries, but current methods are either difficult to scale or not uniform enough to enable MAVEs at the scale of gene families or beyond. We present an improved method called Scalable and Uniform Nicking (SUNi) mutagenesis that combines massive scalability with high uniformity to enable cost-effective MAVEs of gene families and eventually genomes.

## Background

Massive mutagenesis followed by functional assays, commonly known as MAVE or deep mutational scanning, is a powerful strategy for understanding the effects of genetic variation [1–3], dissecting and engineering proteins [4–6], and directed evolution [7]. Modern approaches for generating mutagenesis libraries generally fall into two categories. First, synthesis of oligonucleotides containing programmed mutations followed by subcloning, known as cassette or tile mutagenesis [2,8,9]. Second, synthesis of polymerase chain reaction (PCR) primers containing programmed mutations which bind template DNA and extend to form a mutated strand, followed by various means of degrading and resynthesizing the opposite strand to form mutated double-stranded DNA [10–12].

While cassette mutagenesis yields highly uniform libraries, current DNA synthesis technologies can only yield oligonucleotides up to length ~300, with synthesis quality decaying rapidly with increased length [13]. Since many genes exceed this length, it is necessary to generate sublibraries, which require complex experimental designs that severely limit scalability. The key advantage of primer-based mutagenesis is that it does not have this limitation; in theory, any number of genes of any length can be mutated in a single pot. However, primer-based methods suffer from differences in mutagenesis efficiency between positions, resulting in libraries with highly nonuniform representation of variants [10–12]. Additionally, primer-based methods can generate substantial amounts of wild-type carryover, requiring the use of larger

the Spanish Ministry of Science and Innovation (PID2020-118723GB-I00, EMBL Partnership, Severo Ochoa Centre of Excellence), the la Caixa Research Foundation (LCF/PR/HR21/52410004), the AXA Research Fund, Agencia de Gestio d'Ajuts Universitaris i de Recerca (AGAUR, 2017 SGR 1322), and the CERCA Program/Generalitat de Catalunya. T.L.M. was funded by an EMBO long-term fellowship (ALTF 113-2021). The funders had no role in study design, data collection and analysis, decision to publish, or preparation of the manuscript.

**Competing interests:** The authors have declared that no competing interests exist.

experimental volumes, increased sequencing, and sequencing errors artificially inflating counts for single nucleotide variants [14]. These drawbacks are problematic because they reduce data quality and increase the cost of every step of a MAVE experiment, thereby limiting scalability.

The Atlas of Variant Effects Alliance has the goal to quantify the impact of variation in most human genes and regulatory elements using diverse selection assays [15]. With current rates of progress this endeavor is likely to take decades to achieve [16]. Here we detail a protocol that we term Scalable and Uniform Nicking (SUNi) mutagenesis that represents a two-fold improvement in screening efficiency over the existing state of the art method [12] for variant library construction. SUNi mutagenesis yields highly uniform variant libraries with massive potential scalability.

## Results and discussion

Nicking mutagenesis generates mutated plasmid in four steps: degradation of one DNA strand; annealing and extension of a mutagenic primer; degradation of the opposite strand; and resynthesis of the opposite strand, incorporating the mutation [12] (S1 Fig). Previous data indicated that longer homology arms could improve nicking mutagenesis efficiency [17], and that the melting temperature ($T_m$) of the mutagenic primer was correlated with mutagenesis efficiency [18]. We reasoned that since binding of both homology arms to the template is required for efficient mutagenesis, performance could be improved by $T_m$ optimizing both arms of the primer independently. Therefore, we designed a pool of primers (referred to as opt1) where, for each position, the left and right homology arm had the length between 20–40 nucleotides that had the predicted $T_m$ closest to 61°. These primers were designed to target two 40-codon regions of the μ opioid receptor (MOR) which were chosen because of very high or low GC content (MOR2 = 65.8% GC, MOR6 = 40.8% GC) and so were expected to provide the greatest challenge for the new design. Advances in DNA synthesis have made oligonuceotide pools an affordable, and therefore scalable, option for synthesizing large numbers of sequences. A previous version of nicking mutagenesis synthesized primers as microarray-based oligonucleotide pools, but the quality of these libraries was substantially lower than the original method [18], possibly due to the femtomole-scale yield of microarray synthesis. To maintain the scalability advantage of oligo pools while still maximizing library quality, we synthesized our primers as IDT oPools, which have picomole-scale yield.

The sequential degradation of each DNA strand of a plasmid is accomplished with the nicking activity (cleavage of only one strand of double stranded DNA) of engineered variants of the BbvCI restriction enzyme. We found that some plasmids containing only one BbvCI site are inefficiently digested in the first nicking step, potentially leading to wild-type carryover. Adding a second BbvCI site to the plasmid improved digestion efficiency (S2 Fig). Therefore, we engineered a plasmid bearing MOR to contain two BbvCI sites, and followed the published nicking protocol with minor modifications (S1 Protocol). Sequencing of the mutagenesis libraries revealed similar proportions of programmed mutations (63.8 and 58.8% for opt1 versus 65.3 and 64.2% for standard nicking) and slightly increased wild-type percent (26.9 and 33.2% for opt1 versus 23.8 and 23.3% for standard nicking) but with improved uniformity (log difference (LogDiff) between 90th and 10th percentile of mutants of 0.83 and 0.92 for opt1 libraries versus 1.18 and 0.94 for standard nicking libraries, Fig 1A and 1B). While overall uniformity was improved, there was still substantial positional bias (Fig 1B), which we next sought to understand. However, we found no relationship between mutagenesis frequency (median frequency of all programmed mutations per position) and predicted $T_m$ of left or right mutagenesis primer homology arm, or for minimum, maximum, sum, or difference between left

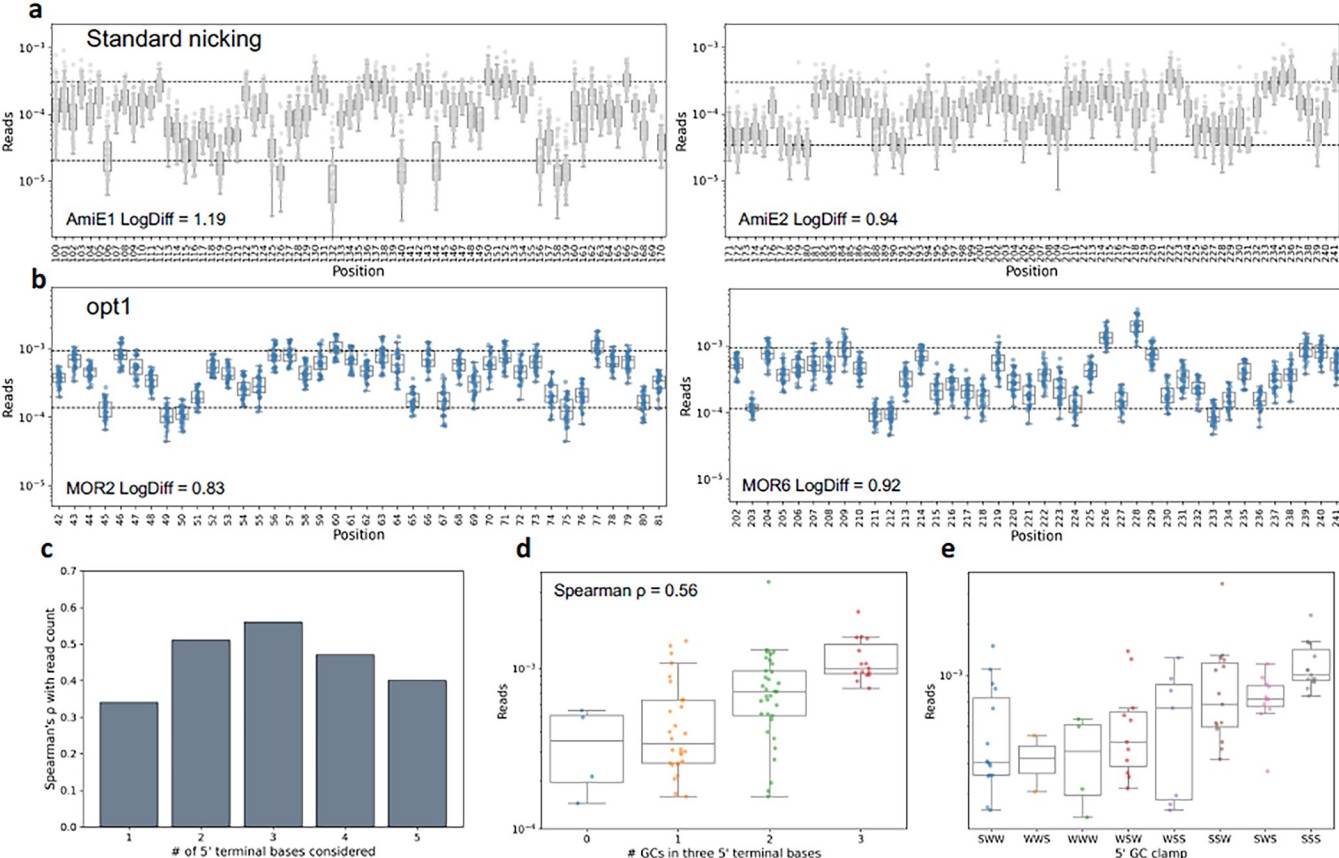

**Fig 1. Optimization and analysis of nicking mutagenesis primer design. a,** Per position mutation frequency presented as fraction of all sequencing reads for standard nicking. Dashed lines indicate 90th and 10th percentile of all mutation frequencies. **b,** Per position mutation frequency presented as fraction of all sequencing reads for opt1 nicking. Dashed lines indicate 90th and 10th percentile of all mutation frequencies. **c,** Spearman correlation between GC content of the 5' terminus and mutagenesis efficiency, when considering between one and five terminal bases. **d,** Mutagenesis frequency of positions with different GC content in the 5' terminal three bases. Spearman $\rho = 0.56$, $p = 6.8\times10^{-8}$. **e,** Mutagenesis frequency of positions with different SW sequences (S = G or C, W = A or T) in the 5' terminal three bases.

and right $T_m$. We also found no contribution of predicted free energy of secondary structure formation of primers (S4 Table).

Surprisingly, we did find a significant contribution of GC content of the five 5' terminal bases of the primer. The strongest signal comes when considering GC content of the three 5' terminal bases (Spearman $\rho = 0.56$, $p = 6.8\times10^{-8}$ Fig 1C and 1D). A GC-rich 3' terminus of a primer (also known as "GC clamp") is widely thought to improve priming efficiency [19], but here we find no contribution of 3' GC clamp (S4 Table). We divided primers based on the 5' terminus sequence and found that primers with SSS, SWS, or SSW sequence (from 5' to 3', where S = G or C and W = A or T) have the highest median mutagenesis efficiency (Fig 1E). Conceptually, the importance of a 5' GC clamp makes sense because the extension step of the mutagenesis PCR is long and at a relatively high temperature (7 minutes at 72°), and if the mutagenic primer terminus is dissociated from the template when the polymerase completes the mutagenic strand, it may polymerize extra bases and make ligation of the mutagenic strand impossible.

We designed a new set of nicking primers (referred to as SUNi), targeting the same regions, and taking advantage of the 5' GC clamp discovery. Briefly, for each position we sought to find a primer that had optimal predicted $T_m$ and also a strong 5' GC clamp (full description in

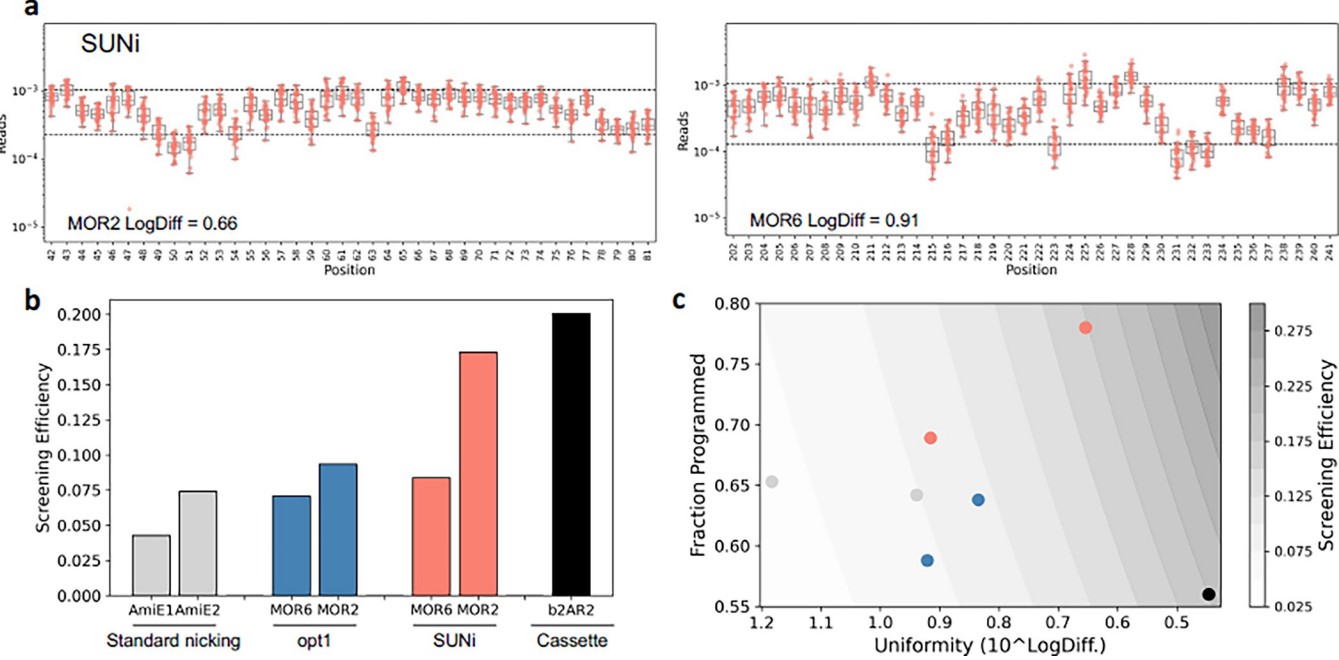

**Fig 2. Performance and comparison of SUNi mutagenesis with other methods. a,** Per position mutation frequency presented as fraction of all sequencing reads for SUNi mutagenesis. Dashed lines indicate 90th and 10th percentile of all mutation frequencies. **b,** Screening efficiency of different mutagenesis methods. **c,** Screening efficiency of different mutagenesis methods, as a function of uniformity and percent programmed. Colors the same as in **b.**

Methods). Further, we reasoned that one contribution to wild-type carryover is NNK primers in which the wild-type codon is encoded by NNK. Since K encodes G and T, for any codon that ends in these bases the wild-type sequence will be present in the NNK pool, and this fully complementary wild-type primer would be expected to outcompete mutation-bearing primers. To minimize this, we used NNK to mutagenize codons that end in A, C, or G, and NNS (S = G or C) if the wild-type codon ended in T. Sequencing of MOR2 and MOR6 SUNi mutagenesis libraries demonstrated increased percentage of programmed mutants (77.5 and 68.9%, respectively, p<0.001 in both cases, Fisher's exact test) decreased percentage of wild-type (13.9 and 23.7%, respectively, p<0.001 in both cases, Fisher's exact test), and improved uniformity (LogDiff = 0.65, 0.92, respectively, p<0.001 in both cases, Kolmogorov–Smirnov two-sided test, Fig 2A).

We wanted to compare methods using a more comprehensive metric, so we calculated screening efficiency $= \frac{\%\text{programmed}}{10^{\text{LogDiff}}}$, a term which incorporates the fraction of programmed sequences in the library and the uniformity of those sequences, which are both important to determine the efficiency of screening the library. Screening efficiency for both libraries increases from opt1 to SUNi designs, and on average SUNi is twice as efficient as the standard nicking protocol (0.128 versus 0.058, respectively, Fig 2B and 2C). We also compared a mutagenesis library made by cassette mutagenesis (b2AR2, 250 nucleotide oligonucleotides introducing mutations at 70 positions). We find that in the best case (MOR2), SUNi screening efficiency approaches that of cassette mutagenesis (0.173 versus 0.200, respectively, Fig 2B and 2C), while requiring substantially less hands-on time and allowing mutagenesis of much larger and many different targets in a single reaction pool. Cassette mutagenesis yields highly uniform libraries (LogDiff = 0.45), but the percent of programmed mutants is low (55.5%, Fig 2C) due primarily to errors in DNA synthesis. We note that while improvements in DNA synthesis

fidelity would improve screening efficiency of cassette mutagenesis libraries, they would remain labor intensive to produce.

We chose to mutagenize regions with high and low GC content, assuming these would be difficult templates for mutagenesis. However, we didn't anticipate the crucial importance of the 5' GC clamp. The data suggests that the mutagenesis efficiency of SUNi is likely related to GC content, indicating that MOR6 is likely difficult while MOR2 is likely an amenable template. We expect SUNi mutagenesis efficiency for regions with intermediate GC content to be intermediate between the examples shown here.

SUNi mutagenesis has the potential to be massively scaled, as there is no theoretical limit to the length of mutated region or the number of mutated regions in a single reaction. The efficiency of screening a SUNi library is twice that of the standard nicking protocol, meaning that at all steps (library generation, screening, and sequencing), the reagents required, and therefore cost, will be halved. We expect SUNi mutagenesis coupled with a panel of selection assays [16] will allow the rapid and cost-effective generation of variant effect atlases for entire gene families. The bright future of MAVEs is reliant on scalable methods for generating high quality variant libraries, and SUNi mutagenesis represents an important step in that direction.

## Conclusions

More efficient libraries empower more scalable experiments that will be necessary for generating atlases of variant effect at the gene-family or genome scale. In this report, we outline design and experimental improvements that improve the screening efficiency of nicking mutagenesis two-fold.

## Materials & methods

### Wet lab protocol

A detailed protocol is available (S1 Protocol) and is very similar to the original nicking protocol [12]. The only difference is in the mutagenic primer annealing and extension step: the original protocol uses three sets of five annealing/extension cycles, spiking in additional primer between the sets. We only use one set of five cycles.

### Opt1 primer design

Primers were designed to introduce all single amino acid mutations and stop codon (via "NNK" codon mutagenesis) for 80 codons in the µ opioid receptor (MOR). To pick a guide for each position, for each homology arm, we found the candidate between 20 and 40 nucleotides with $T_m$ closest to 61˚ (calculated with biopython [20] using the Bio.SeqUtils.MeltingTemp. Tm_NN function). The two pools of opt1 primers were ordered as IDT oPools. Sequences reported in S2 Table.

### SUNi primer design

Like opt1, we designed primers to introduce single amino acid mutations at 80 positions of MOR. For each position, we found the right homology arm in the same way as for Library 1, i.e. the arm between 20 and 40 nucleotides that had predicted $T_m$ closest to 61˚. For the left homology arm, we enumerated all arms that had predicted $T_m$ between 59˚ and 66˚. If one or more of these arms had all three 5' terminal nucleotides as S (degenerate codon notation; S = G or C, W = A or T), the shortest of these was chosen. If there were no SSS 5' termini, then we looked for arms with SSW or SWS termini, and if there were one or more, we chose the shortest arm. If there were no suitable homology arms with SSW or SWS termini, we then

found the arm closest to 64˚ irrespective of 5' terminus. Since we would then predict this primer to be suboptimal, we encoded it twice in the oPool. In this library we used NNK as the degenerate mutagenic codon if the WT codon ended in A, C, or G, but we used NNS if the WT codon ended in T. The two pools of SUNi primers were ordered as IDT oPools. Sequences reported in S2 Table.

### b2AR2 mutagenesis

Oligonucleotides were designed to introduce all possible single amino acid changes, and many double amino acid changes, for a total of 4005 variants. These were synthesized by Twist Bio-science as 250 nucleotide oligos. Q5 High Fidelity polymerase (New England Biolabs) PCR with primers dialout_tile2_[F/R] (primers used in this study reported in S1 Table) and 18 cycles were used to amplify these mutagenic oligos. PCR with primers designed to amplify the rest of the vector besides the region to be mutagenized (b2AR_satmut_tile2_[F/R]) was performed to prepare the vector, and then Gibson assembly was used to introduce the mutagenic oligos.

### Sequencing library preparation

Two stage PCR was performed to amplify each mutated region and append indexed Illumina sequencing adapters. Q5 High Fidelity polymerase (New England Biolabs) was used for all library preparation PCRs. For MOR2 and MOR6 regions, primers MOR_nicking_T[2/6]_seq_[F/R] were used in stage1 PCR to amplify the target and append partial Illumina sequencing adapters, with 50 ng of purified plasmid as template. Cycling protocol was 98˚ for 30s, followed by 17 cycles of [98˚ for 20s, 55˚ for 30s, 72˚ for 30s]. Products were column purified and 0.2% of PCR1 was used as input for PCR2 with primers indexed_i[5/7] and cycled with 98˚ for 30s, followed by 5 cycles of [98˚ for 15s, 64˚ for 30s, 72˚ for 30s]. Products were column purified and sequenced on Illumina Nextseq 500 or Nextseq 2000 instruments. For b2AR2, 10 ng of purified plasmid was used as input to PCR using primers b2AR_Tile2_PCR1_5N_[F/R] and cycling with 98˚ for 30s, followed by 12 cycles of [98˚ for 15s, 66˚ for 30s, 72˚ for 30s]. Products were column cleaned and 0.2% of PCR1 was used as input for PCR2 with primers indexed_i [5/7] and cycled with 98˚ for 30s, followed by 10 cycles of [98˚ for 15s, 64˚ for 30s, 72˚ for 30s]. Products were column cleaned and sequenced on Illumina MiSeq instrument.

### Sequencing data processing

We obtained raw fastq data from the original nicking paper [12] from the Short Read Archive with accession numbers SRR4105481 and SRR4105482. All fastq data were processed identi-cally: first, read pairs were merged and filtered for reads which contained <0.5 expected errors using vsearch [21]. Then, cutadapt [22] was used to trim adapters and only those reads with matching adapters were retained. Variant counts were enumerated by comparing sequencing reads to expected sequences based on mutagenesis strategy (i.e. NNN, NNK, or NNS) and counting only perfect matches. Read processing data in S3 Table.

## Supporting information

**S1 Fig. Schematic overview of the four main steps of nicking mutagenesis.** (PDF)

**S2 Fig. Two BbvCI sites improves digestion efficiency.** Plasmids with either one or two BbvCI sites were digested with Nt.BbvCI, exonuclease I and exonuclease III, per the original

nicking protocol, and digestion products were visualized on an agarose gel.
(PDF)

**S1 Table. Oligonucleotide sequences used in this study.**
(XLSX)

**S2 Table. Opt1 and SUNi mutagenesis primer sequences.**
(XLSX)

**S3 Table. Raw data values for sequencing data processing.**
(XLSX)

**S4 Table. Correlation and p values between opt1 mutagenesis efficiency and all features tested.**
(XLSX)

**S1 Protocol.**
(DOCX)

## Author Contributions

**Conceptualization:** Taylor L. Mighell.

**Data curation:** Taylor L. Mighell.

**Formal analysis:** Taylor L. Mighell.

**Funding acquisition:** Taylor L. Mighell, Ben Lehner.

**Investigation:** Taylor L. Mighell, Ignasi Toledano.

**Methodology:** Taylor L. Mighell, Ignasi Toledano.

**Supervision:** Ben Lehner.

**Writing – original draft:** Taylor L. Mighell.

**Writing – review & editing:** Taylor L. Mighell, Ignasi Toledano, Ben Lehner.

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
