## [Decision Letter · Decision Letter 0]

10 May 2023

PONE-D-23-08204

SUNi mutagenesis: scalable and uniform nicking for efficient generation of variant libraries

PLOS ONE

Dear Dr. Lehner,

Thank you for submitting your manuscript to PLOS ONE. After careful consideration, we feel that it has merit but does not fully meet PLOS ONE’s publication criteria as it currently stands. Therefore, we invite you to submit a revised version of the manuscript that addresses the points raised during the review process.

We look forward to receiving your revised manuscript.

Kind regards,

Paulo Lee Ho, Ph.D.

Academic Editor

PLOS ONE

Journal Requirements:

“This work was funded by a European Research Council (ERC) Advanced grant (883742), the Spanish Ministry of Science and Innovation (PID2020-118723GB-I00, EMBL Partnership, Severo Ochoa Centre of Excellence), the la Caixa Research Foundation (LCF/PR/HR21/52410004), the AXA Research Fund, Agencia de Gestio d’Ajuts Universitaris i de Recerca (AGAUR, 2017 SGR 1322), and the CERCA Program/Generalitat de Catalunya. T.L.M. was funded by an EMBO long-term fellowship (ALTF 113-2021).”

“This work was funded by a European Research Council (ERC) Advanced grant (883742), the Spanish Ministry of Science and Innovation (PID2020-118723GB-I00, EMBL Partnership, Severo Ochoa Centre of Excellence), the la Caixa Research Foundation (LCF/PR/HR21/52410004), the AXA Research Fund, Agencia de Gestio d’Ajuts Universitaris i de Recerca (AGAUR, 2017 SGR 1322), and the CERCA Program/Generalitat de Catalunya. T.L.M. was funded by an EMBO long-term fellowship (ALTF 113-2021). The funders had no role in study design, data collection and analysis, decision to publish, or preparation of the manuscript.”

Reviewers' comments:

Reviewer's Responses to Questions

**Comments to the Author**

1. Is the manuscript technically sound, and do the data support the conclusions?

Reviewer #1: Yes

Reviewer #2: Yes

Reviewer #3: Yes

2. Has the statistical analysis been performed appropriately and rigorously? 

Reviewer #1: Yes

Reviewer #2: Yes

Reviewer #3: No

3. Have the authors made all data underlying the findings in their manuscript fully available?

Reviewer #1: No

Reviewer #2: Yes

Reviewer #3: Yes

4. Is the manuscript presented in an intelligible fashion and written in standard English?

Reviewer #1: Yes

Reviewer #2: Yes

Reviewer #3: Yes

5. Review Comments to the Author

Reviewer #1: Taylor Mighell et al, is a manuscript focused on a method, SUNi, which improves the efficiency of missense mutational library generation. SUNi is a nicking mutation library generation pipeline which includes python code for generating primers and optimized molecular biology protocols yielding libraries with less positional bias compared to other nicking approaches. As nicking and other inverse PCR approaches are amongst the most commonly used library generation pipelines, SUNi will be quite useful to the mutational scanning community. Furthermore, this manuscript, to my knowledge, is the most extensive benchmarking and exploration of nicking mutagenesis which has been technically challenging for some groups to implement efficiently. Overall, the manuscript provides extensive details on why and how improvements were made for nicking mutagenesis. I support the publication of this manuscript with some minor suggestions on how the approach is framed and a few additional technical details.

Major Concerns:

None – overall this study and SUNi is a rigorously done and nicking is convincingly improved compared to the previous state-of-the-art for nicking mutagenesis.

Minor suggestions:

1) In the introduction and throughout the manuscript the authors compare and contrast ‘cassette’ based (synthesized oligo libraries stitched into a gene) with inverse PCR based approaches such as nicking mutagenesis. In this study the authors generated a library for a different gene, B2AR, than the target for SUNi. As such it is hard to directly compare library efficiency as nicking mutagenesis is (as the authors describe and demonstrate well) sensitive to sequence. Furthermore, there have been improvements to the cassette-based approaches which were not implemented in when the B2AR library but have by happenstance been implemented for the entirety of MOR by our group. If the authors wanted they could directly compare these libraries however I would encourage the authors to refocus the narrative and manuscript even more on the (rightful) massive benefits to folks who use nicking mutagenesis and a bit less on the comparison with cassette based approaches as they did not go nearly as deep on cassette-based for the comparison to be satisfying.

2) The authors state throughout the manuscript that in theory nicking mutagenesis provides a benefit for being scalable due to not needing to be assembled in sublibraries. In the manuscript the authors generate libraries using SUNi for two different regions (MOR2 42-81, MOR6 202-241). In reading the text, it was not clear to me (perhaps I misread it) whether these two regions were made in the same pool or in parallel. If they were made in the same pool this should be explicitly stated both in the main text and the methods – that’s awesome. If they were made in parallel than the approach is not so different compared to oligo-based libraries. Overall, it still remains to be seen whether nicking mutagenesis approaches can efficiently make libraries for an entire gene such as MOR in the same pool.

3) Deep mutational scanning ideally allows one to measure how all amino acids impact a genes function at each position. This means for a library generation pipeline such as SUNi one should explore positional bias (called uniformity) as the authors do well but also amino acid bias. Within the manuscript the authors compare the fraction programmed overall. I suggest also comparing how different amino acids are incorporated overall and also per position.

4) Currently the methods section has very little details on the specific molecular biology pipelines used and amounts of reagents which will make implementing SUNi in other labs. I encourage the authors to write a more detailed protocols for the manuscript but even better would be writing out an extremely detailed (perhaps even excruciatingly so) for an open source platform such as Bioprotocols.io to aid others in implementing their approach as some labs have faced considerable challenges in implementing the nicking mutagenesis approaches.

Technical suggestions:

Line 51: The authors state SUNi provides a ‘2-fold improvement’. However it’s unclear as written what is improved 2-fold.

Line 132: Authors state that the percent of programmed mutants is low in cassette mutagenesis – what is the percent? In Fig 2c it looks to be around 0.56. As discussed above it is difficult to compare the two approaches as the target is different.

Line 179: B2AR2 mutagenesis section has very little detail in terms of the number of cycles of PCR and what reagents were used.

My relevant background: My research lab applies genetic screening approaches to study membrane biology and disease. To enable this work we developed two library generation pipelines, SPINE and DIMPLE which use oligo pools or are ‘cassette-based approaches’. As such I am familiar with the molecular biology pipelines but have inherent bias towards ‘cassette-based’ approaches as is represented in my suggestions.

Willow Coyote-Maestas

Reviewer #2: Summary

Taylor Mighell et al, is a manuscript focused on a method, SUNi, which improves the efficiency of missense mutational library generation. SUNi is a nicking mutation library generation pipeline which includes python code for generating primers and optimized molecular biology protocols yielding libraries with less positional bias compared to other nicking approaches. As nicking and other inverse PCR approaches are amongst the most commonly used library generation pipelines, SUNi will be quite useful to the mutational scanning community. Furthermore, this manuscript, to my knowledge, is the most extensive benchmarking and exploration of nicking mutagenesis which has been technically challenging for some groups to implement efficiently. Overall, the manuscript provides extensive details on why and how improvements were made for nicking mutagenesis. I support the publication of this manuscript with some minor suggestions on how the approach is framed and a few additional technical details.

Major Concerns:

None – overall this study and SUNi is a rigorously done and nicking is convincingly improved compared to the previous state-of-the-art for nicking mutagenesis.

Minor suggestions:

1) In the introduction and throughout the manuscript the authors compare and contrast ‘cassette’ based (synthesized oligo libraries stitched into a gene) with inverse PCR based approaches such as nicking mutagenesis. In this study the authors generated a library for a different gene, B2AR, than the target for SUNi. As such it is hard to directly compare library efficiency as nicking mutagenesis is (as the authors describe and demonstrate well) sensitive to sequence. Furthermore, there have been improvements to the cassette-based approaches which were not implemented in when the B2AR library but have by happenstance been implemented for the entirety of MOR by our group. If the authors wanted they could directly compare these libraries however I would encourage the authors to refocus the narrative and manuscript even more on the (rightful) massive benefits to folks who use nicking mutagenesis and a bit less on the comparison with cassette based approaches as they did not go nearly as deep on cassette-based for the comparison to be satisfying.

2) The authors state throughout the manuscript that in theory nicking mutagenesis provides a benefit for being scalable due to not needing to be assembled in sublibraries. In the manuscript the authors generate libraries using SUNi for two different regions (MOR2 42-81, MOR6 202-241). In reading the text, it was not clear to me (perhaps I misread it) whether these two regions were made in the same pool or in parallel. If they were made in the same pool this should be explicitly stated both in the main text and the methods – that’s awesome. If they were made in parallel than the approach is not so different compared to oligo-based libraries. Overall, it still remains to be seen whether nicking mutagenesis approaches can efficiently make libraries for an entire gene such as MOR in the same pool.

3) Deep mutational scanning ideally allows one to measure how all amino acids impact a genes function at each position. This means for a library generation pipeline such as SUNi one should explore positional bias (called uniformity) as the authors do well but also amino acid bias. Within the manuscript the authors compare the fraction programmed overall. I suggest also comparing how different amino acids are incorporated overall and also per position.

4) Currently the methods section has very little details on the specific molecular biology pipelines used and amounts of reagents which will make implementing SUNi in other labs. I encourage the authors to write a more detailed protocols for the manuscript but even better would be writing out an extremely detailed (perhaps even excruciatingly so) for an open source platform such as Bioprotocols.io to aid others in implementing their approach as some labs have faced considerable challenges in implementing the nicking mutagenesis approaches.

Technical suggestions:

Line 51: The authors state SUNi provides a ‘2-fold improvement’. However it’s unclear as written what is improved 2-fold.

Line 132: Authors state that the percent of programmed mutants is low in cassette mutagenesis – what is the percent? In Fig 2c it looks to be around 0.56. As discussed above it is difficult to compare the two approaches as the target is different.

Line 179: B2AR2 mutagenesis section has very little detail in terms of the number of cycles of PCR and what reagents were used.

My relevant background: My research lab applies genetic screening approaches to study membrane biology and disease. To enable this work we developed two library generation pipelines, SPINE and DIMPLE which use oligo pools or are ‘cassette-based approaches’. As such I am familiar with the molecular biology pipelines but have inherent bias towards ‘cassette-based’ approaches as is represented in my suggestions.

Willow Coyote-Maestas

Reviewer #3: Lehner and colleagues present a modification of nicking mutagenesis for the efficient generation of variant libraries. Nicking and PFunkel mutagenesis are templated-based mutagenesis methods for constructions of highly defined mutagenesis libraries; the major issues with nicking mutagenesis are (i.) high wild-type (original plasmid sequence) carryover; and (ii.) lack of library uniformity. The major innovations in this paper are identifying the source of these inefficiencies - the 5' GC content is critically important, and improving the primer sets for mutagenesis decreases the amount of wild-type carryover and improves library uniformity. The paper is well written and figures excellent. I have two minor technical comments.

1. For Lines 117-119, metrics are reported but no statistical significance is attached to the findings. I recommend that the authors compare the distributions by an appropriate statistical test to report the significance of the lower wild-type carryover and library uniformity using their metric defined in the paper.

2. For the GC clamp statement, some reference(s) could be supplied. "A GC-rich 3’ terminus of a primer (also known as “GC clamp”) is widely thought to improve priming efficiency (CITE)".

6. PLOS authors have the option to publish the peer review history of their article (what does this mean?). If published, this will include your full peer review and any attached files.

Reviewer #1: **Yes: **Willow Coyote-Maestas

Reviewer #2: **Yes: **Willow Coyote-Maestas

Reviewer #3: No

---

## [Author Response · Author response to Decision Letter 0]

31 May 2023

Reviewer #1: Taylor Mighell et al, is a manuscript focused on a method, SUNi, which improves the efficiency of missense mutational library generation. SUNi is a nicking mutation library generation pipeline which includes python code for generating primers and optimized molecular biology protocols yielding libraries with less positional bias compared to other nicking approaches. As nicking and other inverse PCR approaches are amongst the most commonly used library generation pipelines, SUNi will be quite useful to the mutational scanning community. Furthermore, this manuscript, to my knowledge, is the most extensive benchmarking and exploration of nicking mutagenesis which has been technically challenging for some groups to implement efficiently. Overall, the manuscript provides extensive details on why and how improvements were made for nicking mutagenesis. I support the publication of this manuscript with some minor suggestions on how the approach is framed and a few additional technical details.

We thank the referee for their enthusiasm and constructive suggestions.

Major Concerns:

None – overall this study and SUNi is a rigorously done and nicking is convincingly improved compared to the previous state-of-the-art for nicking mutagenesis.

Minor suggestions:

1) In the introduction and throughout the manuscript the authors compare and contrast ‘cassette’ based (synthesized oligo libraries stitched into a gene) with inverse PCR based approaches such as nicking mutagenesis. In this study the authors generated a library for a different gene, B2AR, than the target for SUNi. As such it is hard to directly compare library efficiency as nicking mutagenesis is (as the authors describe and demonstrate well) sensitive to sequence. Furthermore, there have been improvements to the cassette-based approaches which were not implemented in when the B2AR library but have by happenstance been implemented for the entirety of MOR by our group. If the authors wanted they could directly compare these libraries however I would encourage the authors to refocus the narrative and manuscript even more on the (rightful) massive benefits to folks who use nicking mutagenesis and a bit less on the comparison with cassette based approaches as they did not go nearly as deep on cassette-based for the comparison to be satisfying.

We appreciate the concern but think that it’s useful to compare SUNi both with original nicking (to show the massive benefits) as well as cassette (to see how SUNi compares with another popular competing method). 

Regarding the fact that we compare mutagenesis of different regions- it is true that ideally we would use the same sequence. However, both of these libraries were constructed for use in DMS experiments, and we sought to make this comparison without using additional resources to construct another library which wouldn’t be used for an experiment. Additionally, as we note in the paper (but now have made clearer), the primary determinant of screening efficiency for the cassette method is oligo synthesis fidelity, which should not vary widely between synthesis templates. The improvements that the reviewer mentions making in their manuscript are related to oligo synthesis quality, therefore we add a note that improved synthesis would indeed improve screening efficiency but would still be labor intensive (changes in italic):

Lines 131-135:

Cassette mutagenesis yields highly uniform libraries (LogDiff = 0.45), but the percent of programmed mutants is low (55.5%, Fig. 2c) due primarily to errors in DNA synthesis. We note that while improvements in DNA synthesis fidelity would improve screening efficiency of cassette mutagenesis libraries, they would remain labor intensive to produce.

It is true that SUNi is sensitive to sequence, however based on the analysis of opt1 libraries, it appears that the critical determinant of SUNi efficiency is GC content (enabling more primers to have “good” GC clamps). Therefore, it is likely that the two MOR segments used in the manuscript (which have low and high GC content) represent reasonable estimates of the SUNi efficiency across a relevant range of GC contents.

2) The authors state throughout the manuscript that in theory nicking mutagenesis provides a benefit for being scalable due to not needing to be assembled in sublibraries. In the manuscript the authors generate libraries using SUNi for two different regions (MOR2 42-81, MOR6 202-241). In reading the text, it was not clear to me (perhaps I misread it) whether these two regions were made in the same pool or in parallel. If they were made in the same pool this should be explicitly stated both in the main text and the methods – that’s awesome. If they were made in parallel than the approach is not so different compared to oligo-based libraries. Overall, it still remains to be seen whether nicking mutagenesis approaches can efficiently make libraries for an entire gene such as MOR in the same pool.

These libraries were made in parallel, simply due to the convenience of being able to directly amplify and sequence with Illumina technology. More recently we have made several full-gene mutagenesis libraries with SUNi and sequenced with PacBio long read platform, showing similar high quality to what was observed for the small regions. We made the decision that it was better to share this method sooner and let people begin using it, as opposed to waiting until we could demonstrate full gene mutagenesis.

3) Deep mutational scanning ideally allows one to measure how all amino acids impact a genes function at each position. This means for a library generation pipeline such as SUNi one should explore positional bias (called uniformity) as the authors do well but also amino acid bias. Within the manuscript the authors compare the fraction programmed overall. I suggest also comparing how different amino acids are incorporated overall and also per position.

It is a good point that per-position as well as within-position uniformity are important determinants of library quality. However, the boxplots in Figures 1 and 2 show the abundance of each unique variant at each position, and it can be seen that the within-position uniformity is much better than the per-position uniformity. Therefore, we focus on improving and reporting on the per-position uniformity.

4) Currently the methods section has very little details on the specific molecular biology pipelines used and amounts of reagents which will make implementing SUNi in other labs. I encourage the authors to write a more detailed protocols for the manuscript but even better would be writing out an extremely detailed (perhaps even excruciatingly so) for an open source platform such as Bioprotocols.io to aid others in implementing their approach as some labs have faced considerable challenges in implementing the nicking mutagenesis approaches.

Thanks for pointing this out. We actually do provide the detailed protocol as a Supplementary Protocol, including amounts of all reagents, but we only called this out in the Results and Discussion section, which could be easy for a reader to miss. In order to ensure that readers see that there is a Supplementary Protocol, we have added a “Wet lab protocol” section as the first section in the Methods.

Lines 160-165:

Wet lab protocol

A detailed protocol is available (Supplementary Protocol 1) and is very similar to the original nicking protocol[12]. The only difference is in the mutagenic primer annealing and extension step: the original protocol uses three sets of five annealing/extension cycles, spiking in additional primer between the sets. We only use one set of five cycles.

Technical suggestions:

Line 51: The authors state SUNi provides a ‘2-fold improvement’. However it’s unclear as written what is improved 2-fold.

We agree that we should specify the improvement and therefore changed the text:

Line 50:

…a two-fold improvement in screening efficiency over the existing state of the art…

Line 132: Authors state that the percent of programmed mutants is low in cassette mutagenesis – what is the percent? In Fig 2c it looks to be around 0.56. As discussed above it is difficult to compare the two approaches as the target is different.

We added the percent programmed to the line indicated and for completeness also added the LogDiff of b2AR:

Lines 133-135:

Cassette mutagenesis yields highly uniform libraries (LogDiff = 0.45), but the percent of programmed mutants is low (55.5%, Fig. 2c) due primarily to errors in DNA synthesis.

We also note that Supplementary Table 3 contains all relevant statistics (including: % programmed, % WT, and % not attributed) for the various libraries used in this study

Line 179: B2AR2 mutagenesis section has very little detail in terms of the number of cycles of PCR and what reagents were used.

Agreed, we added details to this section of the Methods:

Lines 191-193:

Q5 High Fidelity polymerase (New England Biolabs) PCR with primers dialout_tile2_[F/R] (primers used in this study reported in Supplementary Table 1) and 18 cycles were used to amplify these mutagenic oligos.

Based on this change we then also modified the description in the following section (Sequencing library preparation):

Lines 199-200:

Q5 High Fidelity polymerase (New England Biolabs) was used for all library preparation PCRs.

Reviewer #3: Lehner and colleagues present a modification of nicking mutagenesis for the efficient generation of variant libraries. Nicking and PFunkel mutagenesis are templated-based mutagenesis methods for constructions of highly defined mutagenesis libraries; the major issues with nicking mutagenesis are (i.) high wild-type (original plasmid sequence) carryover; and (ii.) lack of library uniformity. The major innovations in this paper are identifying the source of these inefficiencies - the 5' GC content is critically important, and improving the primer sets for mutagenesis decreases the amount of wild-type carryover and improves library uniformity. The paper is well written and figures excellent. I have two minor technical comments.

1. For Lines 117-119, metrics are reported but no statistical significance is attached to the findings. I recommend that the authors compare the distributions by an appropriate statistical test to report the significance of the lower wild-type carryover and library uniformity using their metric defined in the paper.

We performed the appropriate tests and now report the p values in the manuscript:

Lines 117-121:

Sequencing of MOR2 and MOR6 SUNi mutagenesis libraries demonstrated increased percentage of programmed mutants (77.5 and 68.9%, respectively, p<0.001 in both cases, Fisher’s exact test) decreased percentage of wild-type (13.9 and 23.7%, respectively, p<0.001 in both cases, Fisher’s exact test), and improved uniformity (LogDiff = 0.65, 0.92, respectively, p<0.001 in both cases, Kolmogorov–Smirnov two-sided test, Fig. 2a). 

2. For the GC clamp statement, some reference(s) could be supplied. "A GC-rich 3’ terminus of a primer (also known as “GC clamp”) is widely thought to improve priming efficiency (CITE)".

Thanks for the suggestion; in line 99 we added a citation to:

“Molecular cloning: a laboratory manual” by Sambrook et al., 1989.

---

## [Decision Letter · Decision Letter 1]

20 Jun 2023

SUNi mutagenesis: scalable and uniform nicking for efficient generation of variant libraries

PONE-D-23-08204R1

Dear Dr. Lehner,

We’re pleased to inform you that your manuscript has been judged scientifically suitable for publication and will be formally accepted for publication once it meets all outstanding technical requirements.

Kind regards,

Paulo Lee Ho, Ph.D.

Academic Editor

PLOS ONE

Additional Editor Comments (optional):

Reviewers' comments:

Reviewer's Responses to Questions

**Comments to the Author**

1. If the authors have adequately addressed your comments raised in a previous round of review and you feel that this manuscript is now acceptable for publication, you may indicate that here to bypass the “Comments to the Author” section, enter your conflict of interest statement in the “Confidential to Editor” section, and submit your "Accept" recommendation.

Reviewer #1: All comments have been addressed

Reviewer #3: All comments have been addressed

2. Is the manuscript technically sound, and do the data support the conclusions?

Reviewer #1: Yes

Reviewer #3: Yes

3. Has the statistical analysis been performed appropriately and rigorously? 

Reviewer #1: Yes

Reviewer #3: Yes

4. Have the authors made all data underlying the findings in their manuscript fully available?

Reviewer #1: Yes

Reviewer #3: Yes

5. Is the manuscript presented in an intelligible fashion and written in standard English?

Reviewer #1: Yes

Reviewer #3: Yes

6. Review Comments to the Author

Reviewer #1: I am happy with the responses to comments and changes made. My bad missing the detailed methods in the supplement.

Reviewer #3: All technical concerns have been addressed - great work! ********************************************

7. PLOS authors have the option to publish the peer review history of their article (what does this mean?). If published, this will include your full peer review and any attached files.

Reviewer #1: **Yes: **Willow Coyote-Maestas

Reviewer #3: No

---

## [Editor Report · Acceptance letter]

29 Jun 2023

PONE-D-23-08204R1 

SUNi mutagenesis: scalable and uniform nicking for efficient generation of variant libraries 

Dear Dr. Lehner:

I'm pleased to inform you that your manuscript has been deemed suitable for publication in PLOS ONE. Congratulations! Your manuscript is now with our production department. 

Kind regards, 

on behalf of

Dr. Paulo Lee Ho 

Academic Editor

PLOS ONE